# Plastic Reconstruction of Upper Extremity Defects in Necrotizing Soft Tissue Infections

**DOI:** 10.3390/bioengineering12070718

**Published:** 2025-06-30

**Authors:** Karren M. Takamura, Jason J. Yoo

**Affiliations:** 1Department of Orthopaedic Surgery, University of Maryland Shock Trauma, 22 S Greene Street, Baltimore, MD 21201, USA; 2Department of Plastic Surgery, University of Pittsburgh, 3550 Terrace Street, Scaife Hall 6B, Pittsburgh, PA 15219, USA; yoojj@upmc.edu

**Keywords:** necrotizing soft tissue infection, soft tissue reconstruction

## Abstract

Soft tissue reconstruction in patients with upper extremity necrotizing soft tissue infections (NSTIs) can be challenging; these defects can be large with exposed critical structures. Following appropriate source control and debridement, soft tissue reconstruction is based on size, exposed structures, medical co-morbidities and the physiologic status of the patient. There are multiple options for soft tissue coverage from local wound care to free tissue transfer. Dermal substitutes can help prepare a healthy wound bed that can later accept a skin graft. Local rotational flaps, distant pedicled flaps and free flaps are also options depending on the patient and the defect. Patients can have good functional outcomes after soft tissue reconstruction after upper extremity NSTI.

## 1. Introduction

Necrotizing soft tissue infection (NSTI) is a life-threatening condition with high morbidity and mortality rates. This can occur after simple skin lacerations and can progress quickly through the subcutaneous tissue and fascia. Management consists of early detection and diagnosis, administration of antibiotics, early and aggressive surgical debridement to remove the infected and necrotic soft tissues, and critical care support.

Early diagnosis is key in treating NSTIs. A majority of cases exhibit swelling and erythema; however, pain out of proportion is the most consistent finding [1]. Sometimes, it can be difficult to distinguish severe cellulitis from NSTIs. A high index of suspicion should be present when there are bullae, skin ecchymosis, edema beyond the area of erythema and cutaneous anesthesia [2]. Necrotizing soft tissue infections can be classified by the depth of anatomic involvement. Necrotizing dermohypodermitis involves the structures superficial to the deep fascia and muscle and typically has a slower progression [3]. Necrotizing fasciitis involves the deep fascia with rapid progression and causes necrosis of the overlying skin through thrombosis of the perforating vessels [3]. Necrotizing fasciitis can also be further categorized as type I or type II variants. Type I is characterized by polymicrobial infections common in immunocompromised patients, and type II by monomicrobial infections, often from streptococcus pyogenes [4]. Once the diagnosis is made, early surgical debridement is imperative. Delay in surgical treatment is the most important and reliable predictor of mortality [5]. Typically, patients are taken back for multiple debridements until the infection is controlled. In a series of 21 patients with upper extremity necrotizing soft tissue infections, the mean time for grafting and wound resurfacing was 6 weeks (42 +/− 7 days) [6]. Studies have reported an average of two operative debridements prior to coverage [7,8,9,10,11,12]. Oftentimes, there will be large soft tissue defects with varying depth, vascularity and composition [13]. In a study of 99 patients with upper extremity NSTI, the median skin defect was noted to be 110 cm^2^, and most patients with a large surface area of debridement did not require amputation and survived [14]. Coverage options will depend on whether critical structures are exposed, the size and depth of the defect, medical co-morbidities of the patient and the physiologic status of the patient.

Upper extremity NSTIs are less common compared to lower extremity NSTIs, and in a review of upper extremity NSTIs, retrospective studies report mortality rates of 10–36% and amputation rates of 11–38% [12,14,15,16]. Crowe et al. found that vasopressor use outside the operative anesthesia window was the strongest predictor of in-hospital mortality [14]. Amputation may be required in cases despite aggressive debridement and proper antibiotic therapy when source control cannot be obtained. A multidisciplinary approach is imperative, as these patients can present with multisystem organ failure and significant medical co-morbidities. Infectious disease, intensive care, medical subspecialties, anesthesiology, occupational/physical therapy, social work and surgical teams can work together to provide comprehensive care for the patient. This review will coverage options for wounds in patients with upper extremity NSTI.

## 2. Local Wound Care

Local wound care with delayed healing by secondary intention can be considered in cases with small, superficial wounds. In one study, 3 out of 21 patients with upper extremity NSTI underwent healing by secondary intention in small wounds that were less than 4 cm^2^ that only required superficial debridement [6]. Oftentimes, the wounds involved with upper extremity NSTI are larger wounds that can have exposed critical structures, such as tendons, and local wound care is oftentimes not feasible.

## 3. Use of Negative Pressure Wound Therapy (NPWT)

The advent of negative pressure wound therapy (NPWT) has revolutionized the management of open soft tissue wounds. This was developed to increase patient comfort and decrease morbidity, cost and length of hospitalization [17]. Once the wound has been thoroughly debrided of necrotic and non-viable tissue, NPWT can be initiated between repeat debridements, helping to remove excess fluid and debris, decrease bacterial load in the tissue, improve tissue perfusion and promote wound healing [18]. This helps to establish a healthy wound bed, which is essential for further reconstruction. Corona et al. demonstrated the use of NPWT in their series of 20 patients with upper and lower extremity NSTI (40% with upper limb) in 75% of the cases, reporting a 30% mortality rate and four amputations [19]. They concluded that the use of NPWT led to improved wound isolation, reduced nursing time and better patient comfort. Negative pressure wound therapy can also be used in conjunction with dermal substitutes to form granulation tissue, particularly in wound beds with varying depth, which can later be covered with skin grafting or allowed to heal by secondary intention if the wound is small.

Variations in NPWT can also be implemented to aid in infection control after initial surgical debridement. Irrigating NPWT involves antiseptic solutions, such as hypochlorous acid, that can be instilled within the wound bed for a set duration [20]. This exposes the infected wound bed to the antiseptic solution, which is then suctioned out of the wound bed from the negative pressure therapy and repeated at a set time interval. Irrigating NPWT can improve wound healing, reduce the number of debridements and reduce hospitalization duration [21] when used as an adjunct in treating infected wounds [22].

## 4. Delayed Primary Closure

In instances where the surrounding skin is spared with primarily fascial involvement, this skin can be used for delayed primary closure or used in reconstruction. In a study by Mattison et al., in their case series of three patients with NSTI of the upper extremity, one patient did not require skin grafting after NSTI of the right hand and forearm [23]. All of the wounds were closed with local tissue or allowed to heal by secondary intention, with an excellent range of motion of all his joints after discharge.

## 5. Skin Grafting/Dermal Substitutes

Skin grafting is an option if there are no critical structures exposed and there is a healthy bed for skin graft take. Wound matrices such as Integra® Dermal Regeneration Template (IDRT, Integra LifeSciences, Princeton, NJ, USA) or NovoSorb^TM^ Biodegradable Temporising Matrix (BTM) (PolyNovo Biomaterials Pty Ltd., Port Melbourne, VIC, Australia) can be used to prepare the wound bed, as they can help keep the wound bed moist and acts as a dermal replacement layer that will allow for skin graft take (Figure 1a,b). They can also help in wounds with differing contours or deep wounds by replacing lost dermis with a thicker, uniform contour [13]. In addition, the BTM synthetic dermal substitute’s polyurethane layer does not appear to be affected by underlying infection and will continue to integrate with the underlying wound as the purulence drains from the perforations in the seal [13].

Once there is a healthy, vascularized bed, skin grafting can be performed with either a split-thickness autograft or a full-thickness autograft, depending on the size of the area and location of the area that is being covered (Figure 1c–f). A wound vac is helpful in applying uniform pressure on the skin graft, minimizing shear forces, and removing egress through the meshed skin graft [17]. Schwartz et al. reported a 75-year-old female with NSTI of the forearm who underwent complete forearm fasciotomies of the dorsal, volar and mobile wad with exposed tendons upon repeat debridement [24]. This patient underwent staged placement of Integra followed by split-thickness skin grafting with a small area on the dorsal forearm that required local wound care. This patient regained significant function postoperatively with good range of motion and strength.

## 6. Local Regional Flaps

Local rotational flaps can be an option when there is healthy skin nearby that can be mobilized to cover a critical defect. Depending on the location of the defect requiring soft tissue coverage, there are various local regional flaps that can provide coverage within the upper extremity. Special attention should be made to ensure that the donor flap blood supply is sufficiently outside the zone of injury.

For dorsal or volar hand defects, local regional options are primarily derived from the forearm. These include a dorsal metacarpal artery flap, reverse radial forearm flap, posterior interosseous artery flap, and the dorsal ulnar perforator flap. The donor sites can typically be closed primarily or skin grafted. In a case series of upper extremity NSTI by Gonzalez et al., a patient with exposed extensor tendons and bone with a dorsal hand defect, a rotational flap was utilized to cover the defect [8]. Padula et al. described using a dorsal metacarpal artery flap to resurface an exposed thumb interphalangeal joint in a young male with NSTI [6]. In this case, the infection also involved the joint, requiring an open arthrotomy and an eventual fusion. The patient was reported to have good functional recovery at 6 months.

More proximally within the proximal forearm, antecubital fossa or elbow, the donor sites are usually taken from the upper arm. Several fasciocutaneous options include the lateral arm flap, reverse lateral arm flap and reverse medial arm flap [25]. It is often helpful to keep in mind the venous drainage of the reverse flow flaps, as the venous drainage is reversed. Interconnections between veins within the flap can allow for adequate drainage to bypass valves, but more often, maintaining a wide pedicle base and skin bridge provides sufficient drainage through the subdermal plexus. Particularly around joints such as the elbow, the surgeon should pay special attention to the joint position to prevent compression of the venous system and may require postoperative immobilization with splinting.

For smaller elbow defects, pedicle muscle flaps may provide sufficient coverage. A proximally based brachioradialis, flexor carpi ulnaris or extensor carpi ulnaris flap can supply a small amount of vascularized tissue, keeping in mind the functional donor morbidity of sacrificing these muscles. It is also possible for the pedicled latissimus dorsi muscle flap to reach more proximal elbow wounds. This flap is a large muscle that is expendable, has a relatively long pedicle, and can be used to cover large wounds.

In a case report by Na et al., a 72-year-old male with NSTI presented with a large soft tissue defect from the upper arm, crossing the antecubital fossa into the forearm [26]. This large, complex soft tissue defect was covered using multiple reconstructive methods; the medial upper arm and medial elbow were covered with a pedicled latissimus dorsi myocutaneous island flap (20 cm × 9 cm), the cubital fossa was later covered with a full-thickness skin graft to preserve functional motion and the anterior shoulder and volar wrist were initially covered with a MatriDerm graft, followed by a split-thickness skin graft. After 2 months post-operatively, the patient’s flap and grafts were well-healed with a good functional outcome of the arm.

## 7. Remote Pedicled Flaps

Pedicled flaps from the abdomen or groin can be considered in certain cases with exposed critical defects. Benefits of an abdominal- or groin-based flap for reconstruction of the upper extremity are that they are usually outside the area of infection, provide supple soft tissue and do not depend on the blood supply of the upper extremity. Examination of the groin and abdomen is crucial to ensure that there are no prior surgeries or scars that may jeopardize the blood flow to these flaps, such as a history of open inguinal hernia repairs, exploratory laparotomies or open cholecystectomies.

While groin flaps can provide coverage of mid-forearm or distal soft tissue defects of the upper extremity, abdominal-based flaps such as the paraumbilical perforator flap can provide coverage of the mid- to proximal forearm (Figure 2a–d). This flap is based on the periumbilical perforators supplied by the deep inferior epigastric artery [27]. Both the pedicled groin flap and the periumbilical perforator flaps require the upper extremity to remain attached to the trunk, typically for three weeks, while the flap is revascularized by the wound bed of the upper extremity prior to the division of the flap.

Frank et al. described a 52-year-old male with NSTI of the digit that spread proximal to the elbow, which required amputation of the ring finger and radial debridement. A pedicled groin flap was used to cover the exposed tendons, while skin grafting was used to cover the rest of the soft tissue defect, which successfully healed [28]. Balakrishnan et al. reported on a 45-year-old male with a history of insulin-dependent diabetes, recurrent pancreatitis, congestive cardiac failure and anemia, who presented with NTSI of his dominant wrist/forearm, which required debridement of his extensors and radial artery [29]. His defect was covered with a groin flap to cover the dorsal/radial wrist and forearm defect, which was divided at 3 weeks and healed, but lacked extension of his thumb and index fingers.

## 8. Free Tissue Transfers

Free tissue reconstruction may be indicated in certain patients with exposed critical structures that cannot be covered with local or pedicled flaps. However, this can be a difficult reconstruction as patients often present with medically complex conditions with large defects. Optimal timing of the free flap can be difficult to assess; performing a flap too soon could result in an infection under the flap, compromising the viability of the flap. Free tissue transfer may require long operative times, which some patients in critical condition may not tolerate. Donor site morbidity from the harvest flap should also be taken into account, depending on the patient’s soft tissue defect and overall functional status. Special considerations in patients with NSTI include the zone of injury with thrombosed or friable vessels, hypercoagulable states and the surface area of reconstruction. Not infrequently, flaps with long pedicles may be required to get outside the zone of injury or may need extension with arteriovenous loops or interposition vein grafts. However, free tissue has many benefits. Composite flaps can replace lost structures such as tendons or bones. Cutaneous nerves within a flap can be coapted to the sensory nerves of the upper extremity to provide a sensate flap and protective sensation. Lastly, free functional muscle transfer can potentially restore some motion in patients who have lost muscle compartments.

In a study of NSTI patients by Gawaziuk et al., 12 patients who underwent free tissue transfer were compared with 212 patients who did not. Six of the twelve patients had upper extremity NSTI and were all reconstructed using anterolateral thigh free tissue transfers [30]. While all the upper extremity free tissue transfers were successful in the series, there were significant complications, including exposed tendon, partial donor site graft take, seroma, donor site skin necrosis and hematoma in the upper extremities. No functional outcomes were reported in this study.

In a series reported by Kim et al., of the eighteen patients who underwent free tissue transfer for NSTI in all locations, one had a volar forearm defect that was covered with a 30 cm-by-18 cm latissimus dorsi myocutaneous free tissue transfer [18]. The patient was a 62-year-old male with diabetes who was in sepsis with acute kidney injury, and this was complicated by partial flap loss. However, the wounds healed, and there was no recurrence of infection at 15 months post-operation.

## 9. Summary

The treatment of necrotizing soft tissue infections presents a reconstructive challenge. Figure 3 serves as a general algorithm, but each case is unique and different coverage options can provide a good clinical result. As the goal of treatment is aimed at source control via radical debridement, the damage to the soft tissues can be extensive, exposing critical structures such as tendons, bones, nerves and vessels. The reconstructive surgeon has many options, although not every option is suitable. Given the paucity of data on specific reconstructive procedures for these patients, it is difficult to assess the long-term outcomes of these patients. In addition, follow-up on these patients can be difficult [31]. In a systematic review, most patients at an average follow-up period of 18 months either had no or minimal long-term sequelae after upper extremity NSTI [12]. Twenty-four patients (44%) had no loss of function, two (3.7%) had mildly reduced motor function, one (1.9%) had reduced motor and sensory function and one (1.9%) had total loss of motor function. Patients can have good outcomes after soft tissue reconstruction after upper extremity NSTI, and success relies on the careful consideration of the zone of injury, available donor sites, perfusion of the surrounding soft tissues, functional goals and clinical condition of the patient.

## Figures and Tables

**Figure 1 bioengineering-12-00718-f001:**
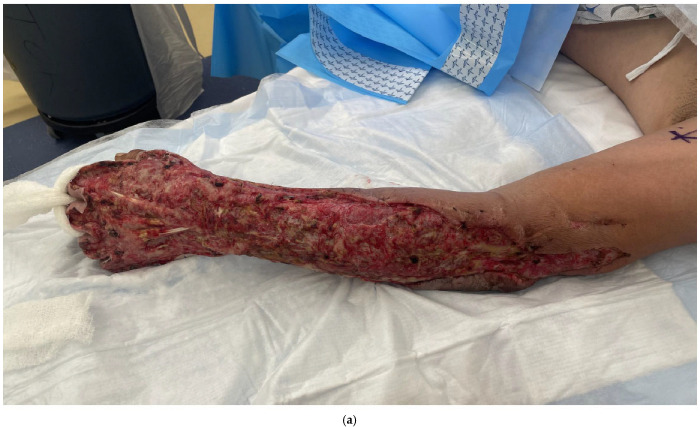
(**a**) A 43-year-old male with necrotizing fasciitis of the left hand, forearm and elbow. (**b**) Wound after multiple debridements and application of Integra, which created a healthy wound bed for skin grafting. (**c**,**d**) After split thickness skin grafting and wound vac application. (**e**,**f**) After the wound graft has healed. At 3 months post-op, he had moderate stiffness of the digits but was otherwise doing well. He was subsequently lost to follow-up and unable to start therapy.

**Figure 2 bioengineering-12-00718-f002:**
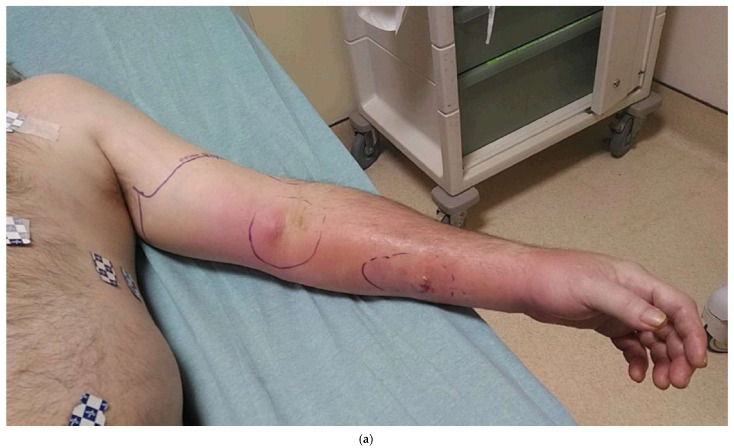
(**a**) A 60-year-old male with necrotizing fasciitis of his left arm. (**b**) After multiple debridements, he had multiple tendons and the distal radius exposed. Given the large defect, a periumbilical perforator flap was chosen. Free flap was considered, but given the distal location of the critical defect and the need to go into the mid-upper arm to get out of the zone of injury for inflow, a pedicled periumbilical perforator flap was chosen. (**c-1**) The critical defect was covered with a pedicled periumbilical perforator flap, and the (**c-2**) remaining exposed non-critical areas were covered with skin allograft to further improve the wound bed for eventual autograft skin grafting. (**d**) The flap was complicated by distal necrosis, but the wound healed with local wound care since there were no exposed critical structures in that area. The rest of the flap and skin graft healed uneventfully.

**Figure 3 bioengineering-12-00718-f003:**
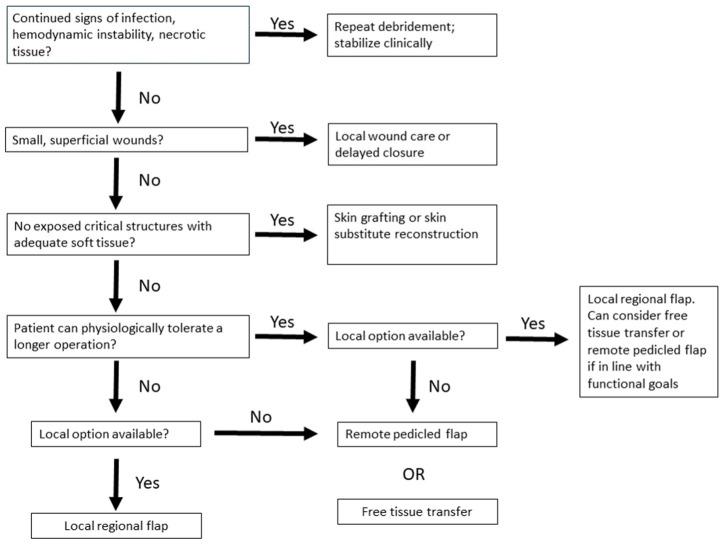
This is a basic algorithm that can be used as a guide for soft tissue coverage in patients with upper extremity NSTI. This is a basic guideline, and each patient’s physiologic status, functional needs and donor site availability should be considered in the soft tissue reconstruction of these patients.

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
