# Peer review of "Plastic Reconstruction of Upper Extremity Defects in Necrotizing Soft Tissue Infections"

_bioengineering, 2025, doi:10.3390/bioengineering12070718_

Round 1

Reviewer 1 Report

Comments and Suggestions for Authors

The article provides a comprehensive contribution to the understanding of plastic reconstruction of upper limb defects in necrotizing soft tissue infections. The organization is clear and the description of therapeutic options is detailed. The work demonstrates familiarity with the current literature.

To further improve the manuscript, I would like to suggest the following points:

Although mortality and amputation rates are mentioned, it would be beneficial to include more quantitative data on success rates, complications, and functional outcomes associated with each specific reconstructive technique.

It may also be useful to add a visual algorithm or flowchart to illustrate the decision-making process for selecting a reconstructive option, considering wound characteristics (such as size, depth, and exposure of critical structures), patient comorbidities, and physiologic status.

A more detailed discussion of long-term functional recovery, covering aspects such as range of motion, strength, sensation, and patient-reported outcomes, would add further value to the manuscript.

Although the interdisciplinary approach is implicit, explicitly emphasizing the role of a multidisciplinary team (including infectious disease specialists, intensive care, and physical therapy) in the comprehensive management of these patients would strengthen this perspective.

Overall, the manuscript has strong scientific merit, providing a clinically relevant and evidence-based review of reconstructive strategies for upper limb NSTIs. These suggestions mainly aim to deepen the quantitative aspects and increase the clarity of the discussion, further reinforcing the contribution of the article to the field.

Author Response

Comment 1: Although mortality and amputation rates are mentioned, it would be beneficial to include more quantitative data on success rates, complications, and functional outcomes associated with each specific reconstructive technique.

Response: While we agree that more quantitative data would be helpful, we were unable to find much information on this since there is such a paucity of data on outcomes after soft tissue reconstruction of NSTI in the upper extremity. We did include any report of outcomes when available (mostly in case reports, we included any information we could find on post-op function or complications). 

Comment 2: It may also be useful to add a visual algorithm or flowchart to illustrate the decision-making process for selecting a reconstructive option, considering wound characteristics (such as size, depth, and exposure of critical structures), patient comorbidities, and physiologic status.

Response: We have added a flow chart as requested, meant to be a general guide. 

Comment 3:  Although the interdisciplinary approach is implicit, explicitly emphasizing the role of a multidisciplinary team (including infectious disease specialists, intensive care, and physical therapy) in the comprehensive management of these patients would strengthen this perspective.

Response: We added the importance of a multidisciplinary team at the end of our introduction

Reviewer 2 Report

Comments and Suggestions for Authors

Sirs,

This manuscript is of practical importance. 

I would suggest some changes, as follows: 

  1. Please, add some detailed information about the types of soft tissue infections. It would be of interest to add a complete classification of soft tissue infections in particular necrotising types as necrotising dermatohypodermitis, fasciitis, ect.
  2. What are the differences among these various types?
  3. In this sense, please explain the differences between phlegmona, cellulitis and necrotising infection. 
  4. I think that this additional information would render the article more compolete and clearer for the reader.  

Author Response

  1. Please, add some detailed information about the types of soft tissue infections. It would be of interest to add a complete classification of soft tissue infections in particular necrotising types as necrotising dermatohypodermitis, fasciitis, ect. We did addend the introduction to include more of the classification of necrotizing soft tissue infections: anatomical differences between dermatohypodermitis and necrotizing fasciitis; type I and II necrotizing fasciitis. 
  2. What are the differences among these various types? Included in point 1. 
  3. In this sense, please explain the differences between phlegmona, cellulitis and necrotising infection.  The main focus of this manuscript is really aimed at reconstruction after necrotizing soft tissue infections and not necessarily the specifics of work up and diagnoses of these infections. Discussing the differences between diagnoses and pathophysiology of phlegmons and cellulitis which do not require reconstruction is a little outside the scope of the manuscript and would distract from the main focus of reconstruction.